# Atmospheric Radiation Parameterization by Neural Ordinary Differential Equations and Related Models

## Abstract

Radiation parameterization schemes are crucial components of weather and climate models, however, they are known to be computationally intensive. Alternatively, they can be emulated with machine learning (ML) regression models. Mainly vertical energy propagation motivates the usage of ML models featuring sequential data processing. We investigate these and related models for radiation parameterization using atmospheric data modeled within an Arctic region. We observe that Neural ODE performs best in predicting both the long- and short-wave heating rates. Furthermore, we substitute the architecture with its discrete form to boost its efficiency while preserving competitive performance. The practical applicability of the models is studied for different model sizes. Finally, we link the trained neural network to the operational weather forecast model and assessed its performance versus the conventional radiation parameterization. We receive a speedup of 26.5 times of the radiation steps without significant loss of accuracy. The proposed parameterization emulator dramatically reduces the computational burden and the carbon footprint of weather forecasting.

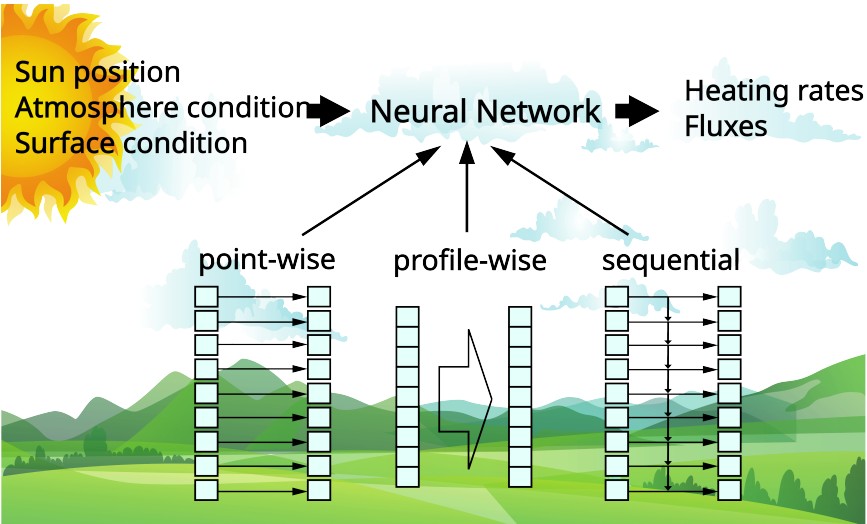

Figure 1: Three types of architectures based on their approach for treating spatial information: point-wise, profile-wise, and sequential

## 1 Introduction

The role of numerical weather prediction (NWP) cannot be overestimated. It is essential for numerous aspects of our daily life and critical systems: public safety, agriculture, aviation and transportation, energy, climate change, making it a cornerstone of our modern society.

Incorporating atmospheric radiative transfer into NWP models via a radiation parameterization is critical for simulating atmospheric dynamics. However, from a computational perspective, radiation calculations present unique challenges due to their significantly higher computational cost in comparison with other parameterizations.

At the core of these calculations are radiative transfer models, which require solving complex linear transport integro-differential equations (Liou, 2002). These solutions must be integrated over multiple dimensions—height, angles, and frequencies—further increasing the computational load. A major bottleneck stems from the highly frequency-dependent nature of radiation absorption by gases, which makes the modeling process inherently complex. Additionally, scattering and absorption by aerosols introduce further layers of difficulty.

The traditional approach to simulating electromagnetic propagation through the atmosphere is the line-by-line radiative transfer model (LBLRTM) (Clough et al., 1992), which meticulously tracks thousands of individual absorption lines. While highly accurate, LBLRTM's computational demands make it infeasible for large-scale, real-time applications.

To address this, the rapid radiative transfer model for global climate models (RRTMG) (Iacono et al., 2008) applies algorithmic optimizations that dramatically reduce computation time without sacrificing accuracy. Yet, despite these advances, radiative transfer remains one of the most computationally expensive components in weather modeling.

Given some data is available, we can emulate a radiative transfer with a regression problem: predict heating rates and radiation fluxes based on the Sun's position, surface, and atmosphere conditions (temperature, pressure, humidity, and others). A variety of neural network (NN) architectures were successfully applied to this regression problem (Chevallier et al., 1998; Krasnopolsky et al., 2010). Recent works employed feedforward, convolutional networks, and transformers (Pal et al., 2019; Liu et al., 2020; Song & Roh, 2021; Lagerquist et al., 2021; Yao et al., 2023) and recurrent networks (Ukkonen, 2022; Yavich et al., 2024).

Within the atmosphere, electromagnetic radiation mainly travels in the vertical direction. Multiple reflections can be considered as some special dynamics of establishing final heating profile. This motivates to approximate the transfer with an ordinary differential equation (ODE) with an unknown right-hand side. After the right-hand side is estimated, the ODE can be solved to predict heating rates. This motivates to apply neural ordinary differential equations (Neural ODEs, NODE) to this problem. Neural ODEs offer a powerful framework for modeling dynamic systems by means of machine learning (Chen et al., 2018). It models the data transformation through a continuous dynamic equation instead of discrete layers as in traditional neural networks.

In this paper, we explore the possibilities of Neural ODEs for atmospheric radiative transfer. Our experiments involve atmospheric data modeled with WRF (Skamarock et al., 2019) within an Arctic region. We focused our work on the Arctic region because of its high economic potential, see e.g. Li et al. (2023), and complex climate.

Atmospheric radiative heating is considered as an adiabatic process in weather forecast and climate modelling. Radiation transfer itself is described at the quantum level and involves numerous transfers, scatterings and reflections. We found that an iterative profile-wise architecture for a Neural ODE is necessary to emulate radiative flux propagation and the underlying physics. On the other hand, sequential and shallow networks architectures emulate radiation transfer in a single direction only, therefore are inferior, which is supported by our experiments.

To sum up, our main contributions are the following:

1. We investigated Neural ODE and related architectures for application in atmospheric radiative transfer;

2. We find that the optimal architecture is the profile-wise RNN, the discrete substitution of the Neural ODE;

3. We linked the trained neural network to WRF operational weather forecast model and assessed its performance versus the conventional radiation parameterization. We received a speedup of 26.5 times of the radiation steps.

## 2 EMULATION OF RADIATIVE TRANSFER

### 2.1 REGRESSION PROBLEM

The search for an optimal approach that would balance both accuracy and efficiency in calculation of the radiative heating rate is still an open question. If a modeling domain and vertical resolution are fixed, we can think of these calculations as a regression problem, reusing earlier data. The input data would be physical fields that describe particular atmospheric and earth surface conditions as well as the Sun's position. The output data would be the heating rate. Such an approach would work with broadband variables directly and thus avoid multiple tedious numerical integrations and differentiations. Alternatively, upward and downward fluxes can be predicted with regression models either within the whole vertical profile or at the Earth's surface and top of the atmosphere only.

Notice the heating rates due to SW and LW radiation can be predicted simultaneously with a multi-output regression model or separately with a set of single-output regression models.

### 2.2 BASIC ARCHITECTURES

All architectures used in our research can be briefly classified into three types based on their approach to treating spatial information: point-wise, profile-wise, and sequential, Figure 1.

Point-wise type uses only local information, the features evaluated at the given height and location to predict the target at the same point. In such an approach the problem is treated as a canonical tabular regression, and gradient boosting like CatBoost (Dorogush et al., 2018) can be used as a simple baseline.

Profile-wise architectures predict the whole target profile at the given location at once and necessitate the whole profile of input features. Commonly, the profile is discretized by the height and convolutional neural networks (CNN) are utilized to effectively treat the spatial dependencies.

The sequential type is the most relevant in terms of physicality as the implicit dynamics describe the vertical transfer process. These architectures parameterize the transition model along the height or pressure gradient, and input is assimilated at each layer sequentially. As a baseline of this approach, we used the gated recurrent unit (GRU) model (Chung et al., 2014).

### 2.3 NEURAL ODE

The basis of Neural ODEs, following (Chen et al., 2018), is represented by the differential equation:

$$\frac{d\mathbf{x}(t)}{dt} = f(t, \mathbf{x}(t), \theta), \tag{1}$$

where $\mathbf{x}(t)$ represents the state of the system at time $t$, $f$ is a neural network with parameters $\theta$ that defines the system's dynamics. Usually, the MLP or CNN architectures are utilized as the parameterization of $f$. The latent state $\mathbf{x}(t)$ has constant dimension, so the models input is projected linearly into the chosen space to form an initial state $\mathbf{x}(t_0)$, and the output is a projected linearly from final state $\mathbf{x}(t_1)$, which is computed through integration.

Numerical methods for integrating differential equations, such as Euler methods, Runge-Kutta methods, and adaptive methods that dynamically choose the step size to control the solution convergence, are used to solve Neural ODEs.

Neural ODEs are trained through backpropagation, the gradients can be computed either directly through the dynamic equation using automatic differentiation or more memory-efficiently through the adjoint method. The adjoint method treats gradients as solutions of a reverse-time differential equation, integrating it backward in time. However, naive implementations of the adjoint method suffer from inaccuracy in reverse-time trajectory (Zhuang et al., 2021). In our work, we have used a specific implementation called MALI (Zhuang et al., 2021) that increases accuracy in gradient estimation.

## 2.4 Substituting Neural ODE

While the adjoint method makes possible the training of Neural ODE with an arbitrarily accurate black-box solver, the inference time is still inferior to that of feed-forward neural networks. Moreover, the inference time can vary from sample to sample in the case of adaptive solvers like MALI. While such behavior is crucial for accurate integration, it was shown recently that the neural networks can balance the impurities of simpler solvers to the level when several steps of the Euler schema are enough (Ivanov & Ailuro, 2024). To be specific,

$$\mathbf{x}(t + \Delta t) \approx \mathbf{x}(t) + f(t, \mathbf{x}(t), \theta)\Delta t, \tag{2}$$

where $\Delta t = \frac{1}{p}$ and $p$ is number of integration steps. When $\Delta t$ approaches zero, the Neural ODE dynamics 1 is reproduced. When $p$ is considerable small ($p \approx 10$), the Neural ODE model effectively turns into a recurrent neural network, where the single layer is reapplied several times to the hidden state without assimilation of new inputs, or into a residual network with weights shared among all layers:

$$\mathbf{x}_{t+1} = \mathbf{x}_t + \frac{1}{p}f(t, \mathbf{x}_t, \theta) = g(t, \mathbf{x}_t, \theta); \quad \mathbf{x}_p = g^p \circ \mathbf{x}_0 \tag{3}$$

The number of 'integration' steps $p$ should be fixed and can be treated as a hyperparameter. Also automatic gradients for backpropagation is sufficient, therefore the model training is more stable. Notably, in the case of the profile-wise approach with convolutions in the approximation of the right-hand side $f$, the number of 'integration' steps defines also the receptive field of the model.

To the aid of simplification, we will refer to this architecture as a profile-wise convolutional RNN or just RNN.

## 2.5 Liquid Neural Network

Liquid time-constant networks control the time-dependency to emulate biological neurons. They treat the system as a specific form of Neural ODE:

$$\frac{d\mathbf{x}(t)}{dt} = -[w_\tau + f(\mathbf{x}(t), I(t), \theta)] \cdot \mathbf{x}(t) + A \cdot f(\mathbf{x}(t), I(t), \theta), \tag{4}$$

where $\mathbf{x}(t)$ is latent state, $I(t)$ is an input, $w_\tau$ is a time-constant parameter vector, and $A$ is a bias vector. For such formulation, the approximation of a solution was introduced, (Hasani et al., 2022):

$$\mathbf{x}(t) = B \cdot e^{-[w_\tau + f(\mathbf{x}, I, \theta)]t} \cdot f(-\mathbf{x}, -I, \theta) + A \tag{5}$$

Authors suggested a modification named a closed-form continuous-time neural network model (CfC):

$$\mathbf{x}(t) = \sigma(-f(\mathbf{x}, I, \theta_f)t) \cdot g(\mathbf{x}, I, \theta_g) + [1 - \sigma(-f(\mathbf{x}, I, \theta_f)t)] \cdot h(\mathbf{x}, I, \theta_h) \tag{6}$$

where $g$ and $h$ are feed-forward neural network. In the used setting $f$, $g$ and $h$ are two-layer perceptrons with the shared first layer, and the variable $t$ is the height over the surface rather than time. The CfC method achieves computational efficacy while still offering a sequential dynamic approach with physical origin.

## 2.6 Other Architectures

Inspired by the perturbation theory, commonly utilized in physics, we explored polynomial-based kernels for neural networks instead of perception-like kernels with rectified linear units. By the replacement of the activations to polynomials we have turned CNN into a Polynomial network (PNN), Neural ODE into Polynomial ODE (PODE), and RNN into Taylor Mapped Polynomial Neural Networks (TMPNN) (Ivanov & Ailuro, 2024).

In the process of architecture search, we also experimented with ODE-base modification of gated recurrent unit (ODEGRU) (Zhao et al., 2023). The model was used in the sequential regime.

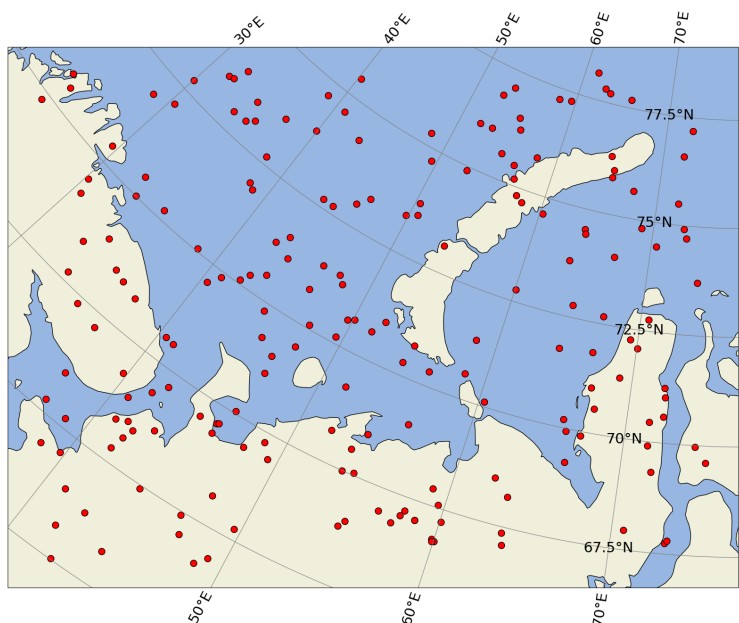

Figure 2: Modeling domain. The red disks indicate vertical profiles and surface data kept on 2015-06-25

## 3   DATA AND DATA PROCESSING

As a dataset, we used a part (2015–2016 years) of the long-term atmospheric hindcast of the Pechora Sea region, Fig 2, kindly conveyed to us by the authors of Yavich et al. (2024). Such a region was used because of the presence of various weather and surface conditions: polar days and nights, a large number of cloudy days, sea and land surface often covered by snow and ice in winter. The horizontal resolution was 3 km, while the temporal resolution was 10 s. The spatial grid dimensions were $N_x = 422$, $N_y = 562$, and $N_z = 49$.

Model prognostic, diagnostic, and radiation variables were saved every hour. Specifically, the following surface variables were saved: cosine of the solar zenith angle, COSZEN, solar constant, SOLCON, surface albedo, ALBEDO, longitude, as well as the following spatial variables: specific humidity, SH, cloud fraction, CF, cloud and ice mixing ratios, QCLOUD and QICE. Further total pressure (P_TOT) and absolute temperature (T_KEL), were calculated and saved. Ozone temporal and spatial mixing ratio distribution OZONE was prepared. The RRTMG radiation parameterization was used to calculate uncoupled potential temperature tendencies due to shortwave (SW) and longwave radiations (LW), RTHRATSW and RTHRATLW, respectively. Both potential temperature and potential temperature tendency were transformed to absolute temperature and absolute temperature tendency, respectively. All the predictors and targets are listed in Table 1. We thus had six scalar and six sequential input variables and two target sequential variables.

Training and testing data involved uniformly distributed data within a year: 350 days for testing (essentially all the days of 2015) and 91 days for training (every fourth day of 2016). Every chosen day included data within 24 hours (sampled every hour) and at 100 random surface locations, Fig. 2. This type of data filtration made the dataset computationally feasible and removed redundant information, while keeping sufficient representation of day/night and seasonal changes. We kept 2015 data (840'000 profiles overall) for learning and 2016 data (218'000 profiles overall) for testing.

Table 1: Selected predictor and target variables. Predictors are selected among other available features by its importance quantification provided by the CatBoost model.

| 2D Predictors | 3D Predictors | Targets |
|---|---|---|
| XLAND | SH | RTHRATSW |
| COSZEN | P_TOT | RTHRATLW |
| SOLCON | T_KEL | |
| ALBEDO | QCLOUD | |
| EMISS | QICE | |
| TSK | OZONE | |

## 4 EXPERIMENTS

### 4.1 IMPLEMENTATION

Almost all models are implemented in PyTorch and trained from scratch with skorch (Tietz et al., 2017). The initial learning rate was set to $10^{-2}$, while other parameters were default. Each model is trained for 100 epochs. For training CatBoost and TMPNN we used their standard implementations. To stabilize training, all inputs are normalized to have the minimum value of zero and maximum value of one, while the target variables are scaled to zero mean and unit variation.

Several models were investigated with different sizes, therefore we mention them with the number of dimensions in the hidden space, for example, CNN 128 stands for the convolutional network with 128 channels. For substituted neural odes we also tried several values of the number of integration steps, e.g. RNN 128,7 stands for Neural ODE with seven steps in the Euler scheme and a convolutional layer with 128 channels.

All convolutional networks are parameterized by two-layer kernel with the use of rectified linear unit activation (ReLU), while all polynomial models consist of one convolutional layer and one polynomial-generation layer.

### 4.2 ACCURACY AND INFERENCE TIME ASSESSMENTS

The detailed RMSE metrics per each target and inference time are presented in Table 2 for the single-output regression task. In terms of inference time, the multi-output task is preferable as all targets are predicted simultaneously, however, the effective representation should be learned, the models performance for multi-output regression is presented in Table 3. The inference time is provided with optimal batch size for each model, the tests have been run on GPU. The number of millions floating-point operations (MFLOPs) is provided for the single input (batch size one). We also compare results with according to bidirectional RNN (RNN_bi) reported in the literature on the same data (Yavich et al., 2024).

Generally, the superior performance is achieved by profile-wise type architectures. The best LW RMSE of 0.0936 K/d is achieved by Neural ODE architecture in single-output formulation, while the best SW RMSE of 0.0200 K/d is achieved by RNN architecture in multi-output regime. Among the sequential models, CfC is the only one capable to learn long-wave heating rate profile. We provide examples of generated profiles for RNN and CfC in Appendix A, Figures 7, 8. Appendix B presents R2 skills of the studied models. For our most accurate models it reached 99.7%.

Table 4 compares SW and LW heating rate RMSE across different recent works. We observe, that our result is superior to other works. Yet we have to emphasize RMSE depends not only in the machine learning model, data trasformation, and selected predictor list, but also on the region and atmosphere model. That is why a direct comparison of the numbers in Table 4 is injudicious.

For further investigation we chose multi-output profile-wise RNN 16,5 as the optimal model in terms of maximizing both fidelity and operational speed. Its error in dependence of the height is depicted in Figure 3. Notably, the models performance declines near the edges of profiles (near the surface and the top of the atmosphere) and at the bottom of the stratosphere.

Table 2: Single-Output performance. Targets are predicted by separately trained networks. For prediction time of both targets the reported time should be twiced. Dash sign marks runs where models have failed to converge. Numbers after model names are hyperparameters: the first one governs the dimension of the hidden state, the second number is the number of steps in RNN-based architectures.

| Type | Model | Test SW RMSE (K/d) | Test LW RMSE (K/d) | Prediction Time ($10^{-5}$s) |
|------|-------|--------------------|--------------------|------------------------------|
| Point-wise | CatBoost | 0.0944 | 0.3886 | 0.3 |
| | NODE | 0.1580 | - | 77 |
| | TMPNN | 0.0934 | 0.3016 | 0.8 |
| Profile-wise | CNN 128 | 0.0606 | 0.1365 | 1.3 |
| | CNN 256 | 0.0568 | 0.1285 | 1.3 |
| | RNN 128,7 | 0.0327 | 0.1125 | 1.4 |
| | RNN 256,7 | 0.0321 | 0.0974 | 1.9 |
| | RNN 512,10 | 0.0305 | 0.0950 | 3.7 |
| | NODE 128 | 0.0372 | 0.0992 | 2.2 |
| | NODE 256 | 0.0345 | 0.0979 | 3.1 |
| | NODE 512 | 0.0305 | **0.0936** | 5.8 |
| | PNN 2 | 0.1408 | - | 1.3 |
| | PNN 4 | 0.1091 | - | 4.2 |
| | TMPNN 16,2 | 0.0620 | 0.1701 | 1.3 |
| | TMPNN 16,7 | 0.0380 | 0.1526 | 1.4 |
| | PODE | 0.0372 | 0.1761 | 1.8 |
| Sequential | GRU | 0.0563 | - | 1.9 |
| | ODEGRU | 0.0614 | - | 2.1 |
| | CfC | 0.0251 | 0.2113 | 1.3 |

Table 3: Multi-Output performance. Targets are predicted by one network simultaneously. The standard deviations are estimated by training models from three random initializations. Numbers after model names are hyperparameters: the first one governs the dimension of the hidden state, the second number is a number of steps in profile-wise RNN. The models were trained for 100 epochs. $^{\dagger}$ indicates that the model was trained for 1000 epochs.

| Type | Model | Test SW RMSE (K/d) | Test LW RMSE (K/d) | Prediction Time ($10^{-5}$s) | MFLOPs |
|------|-------|--------------------|--------------------|------------------------------|--------|
| Profile-wise | RNN$^{\dagger}$ 16,5 | 0.0252±0.0002 | 0.1110±0.0010 | 1.2 | 0.87 |
| | RNN 64,3 | 0.0261±0.0004 | 0.1114±0.0014 | 1.3 | 7.7 |
| | RNN 128,7 | 0.0230±0.0002 | 0.1068±0.0012 | 1.4 | 68.4 |
| | RNN 512,10 | **0.0200±0.0020** | 0.1010±0.0020 | 6.6 | 1550 |
| | NODE 512 | 0.0255±0.0073 | 0.1057±0.0034 | 14 | 2780 |
| Sequential | CfC | 0.0273±0.0002 | 0.1965±0.0058 | 1.4 | 8.3 |
| | RNN_bi | 0.0425 | 0.25 | 2.5 | 9.9 |

## 4.3 PERFORMANCE WITHIN THE OPERATIONAL WEATHER FORECAST MODEL

We linked the trained profile-wise RNN to the WRF operational Weather Forecast. The WRF is implemented mainly in Fortran, we thus used the C++ frontend to PyTorch to link our trained model (see Figure 4). A run with neural parameterization was compared with a run with the standard RRTMG parameterization. The starting times were 1 July 2020 at 12:00 UTC. The computational grid was 280 × 210 × 49, making the horizontal resolution 6 km. The time step was 20 seconds, while the radiation module was called every 6 minutes. The runs were performed on a single CPU.

|  | SW | LW |
|---|---|---|
| This work, RNN 16,5 | 0.025 | 0.111 |
| Yao et al. (2023) | 0.032 | 0.139 |
| Yavich et al. (2024) | 0.042 | 0.250 |
| Ukkonen (2022) | 0.160 | n/a |

Table 4: Comparison of SW and LW heating rates RMSE across different works.

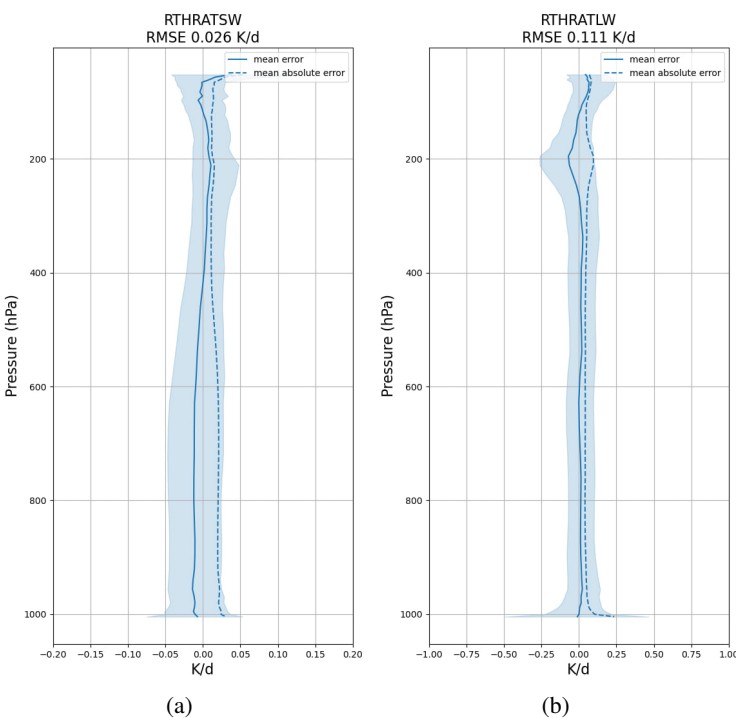

(a)                                                    (b)

Figure 3: Test error distribution versus mean level pressure for profile-wise RNN. The solid and dotted curves indicate the mean error and mean absolute error respectively. The filled area indicates the range from 5th to 95th percentile of the error.

Figure 6 illustrates long-wave radiation fluxes after 1 hour of modeling, i.e. at 13:00 UTC. We observe that the fluxes match fairly well. The neural network managed to emulate all the main features correctly, e.g. larger radiation over the land surface and smaller over the sea.

Figure 5a) shows the mean air temperature near the surface within one week. We see that the temperatures differ by at most $1°C$, which is appropriate for the operation of the weather forecast.

In Figure 5b), we show the CPU time of a single WRF time step for both parameterizations.

The flat parts of the plots on 5b) corresponds to the regular time-steps, i.e. steps on which the radiation module is not called, thus they are equal for both runs, near 9.6 s. The peaks, 106.8 s for RRTMG and 13.3 s for RNN, correspond to the radiation module CPU time. Subtracting regular time-steps CPU time, we find the radiation module calls took 97.2 s and 3.7 s, respectively. This means that we achieved a remarkable speedup of 26.5 times. Overall, one-hour WRF modelling time was reduced from 2700 s to 1765 s to, i.e. in 1.5 times.

## 5   CONCLUSION

In this work, we assessed the performance of short- and long-wave radiation transfer emulation via Neural ODEs and related models. We conducted our experiments using hindcast data from the Pechora Sea region.

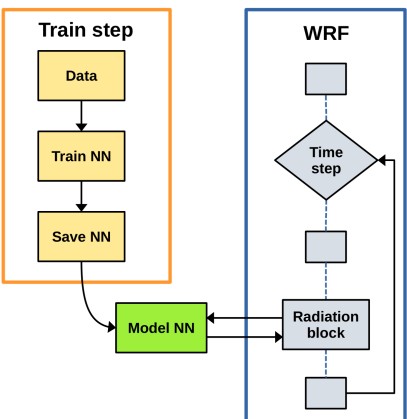

Figure 4: Linking scheme of the trained profile-wise RNN to the WRF

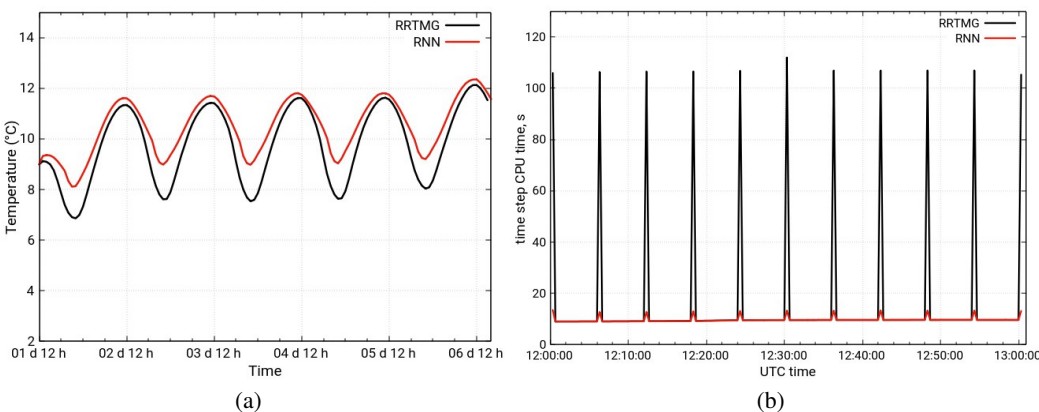

(a)        (b)

Figure 5: (a) WRF temperature at 2 m within a week modelled with RRTMG and profile-wise RNN 16,5 radiation parameterizations. (b) WRF time steps CPU time for RNN and RRTMG parameterizations.

We found that an iterative profile-wise architecture for a Neural ODE is necessary to emulate radiative flux propagation and the underlying physics. On the other hand, sequential and shallow networks architectures emulate radiation transfer in a single direction only, therefore are inferior, which is supported by our experiments.

Overall, 25 models of 11 architectures and three types were tested with this dataset. The superior results are achieved by profile-wise models, especially by Neural ODE and its discrete substitution, which we have chosen as an optimal model in terms of fidelity and operational speed. We suppose that the iterative nature of the proposed parameterization mimics the adiabatic process of numerous radiance reflections at each level of the atmosphere.

We also received preliminary results on the use of the designed and trained neural ODE substitution within operational weather forecast. We verified heating rates and temperatures and observed that they match fairly well. The parameterization module was accelerated in 26.5 times, without significant decrease of accuracy. We plan to conduct more extensive analyses on the neural parameterization performance within the WRF in our future work, e.g. performance during severe weather events and generalization over other regions.

WRF is a regional atmospheric model thus requiring reconfiguration for every new region. Similarily, our parameterization emulator can be successfully trained for a particular region and used to speedup the radiation module within operational weather forecast (where speed is critical) or hindcast modelling (to save compositional resources).

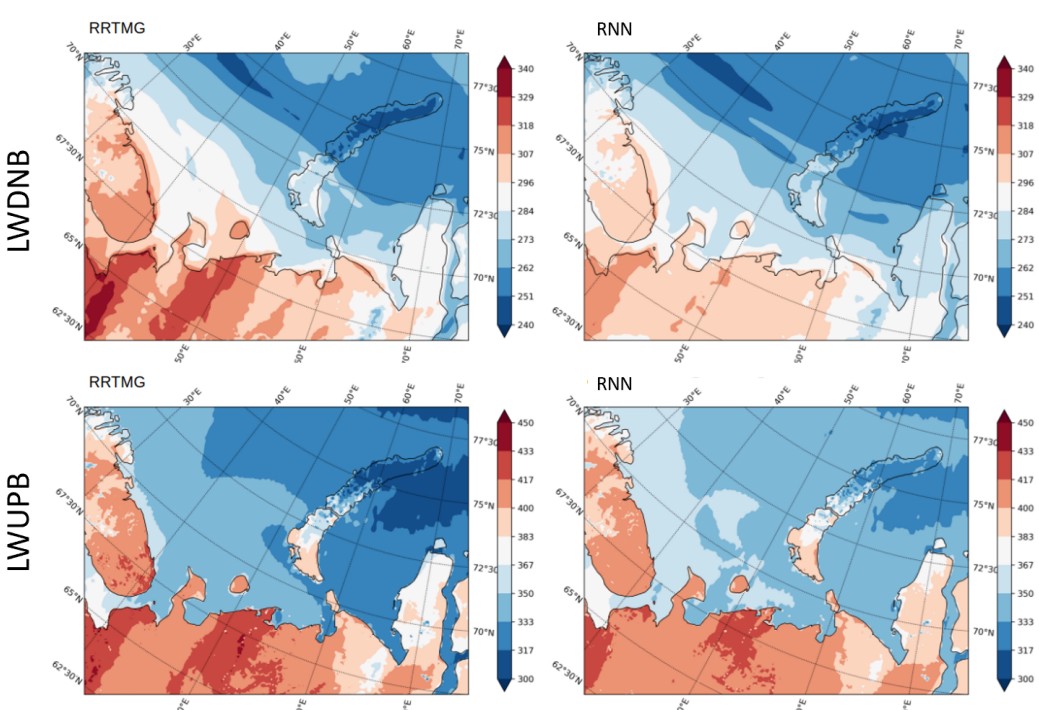

Figure 6: Long-wave radiation fluxes after 1 hour of modeling at 13:00 UTC July 01, 2020.

Another attractive direction of future research is to apply Neural ODEs to other complex parameterization: boundary layer parameterization, sub-grid turbulence, and parameterizations arising in sea ice modeling.

## 6 REPRODUCIBILITY

The developed source code was attached to the manuscript as a supplementary material. The dataset is available upon written request.

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

APPENDIX

## A  PROFILE EXAMPLES

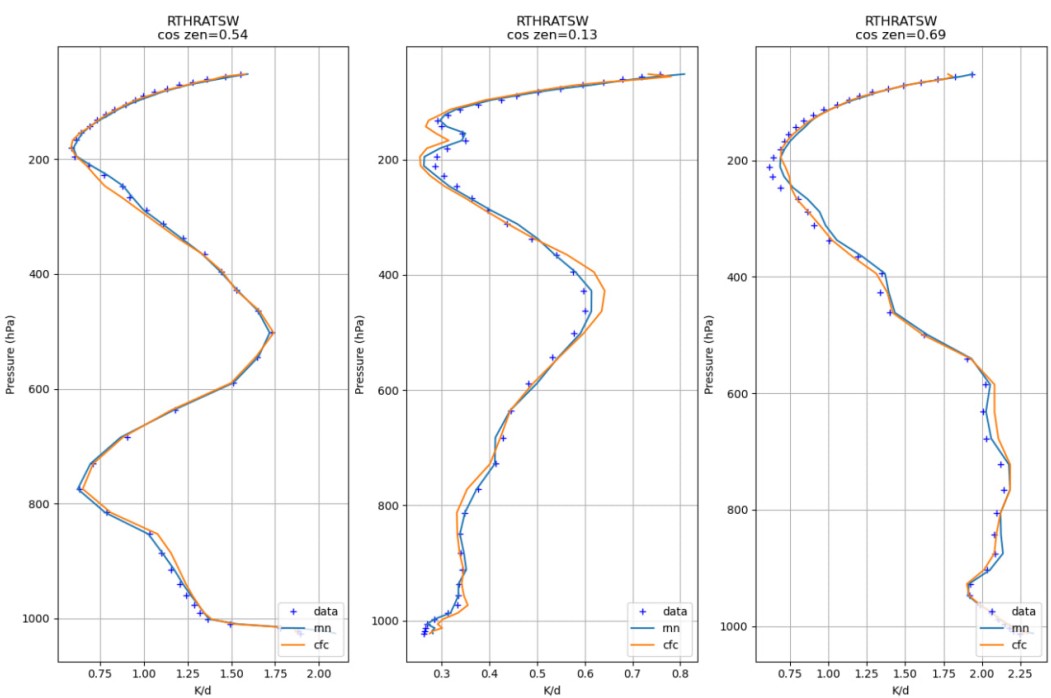

Figure 7: Exemplary short–wave profiles predicted by profile-wise RNN (blue line) and sequential CfC (orange line) models versus RRTMG data (crosses).

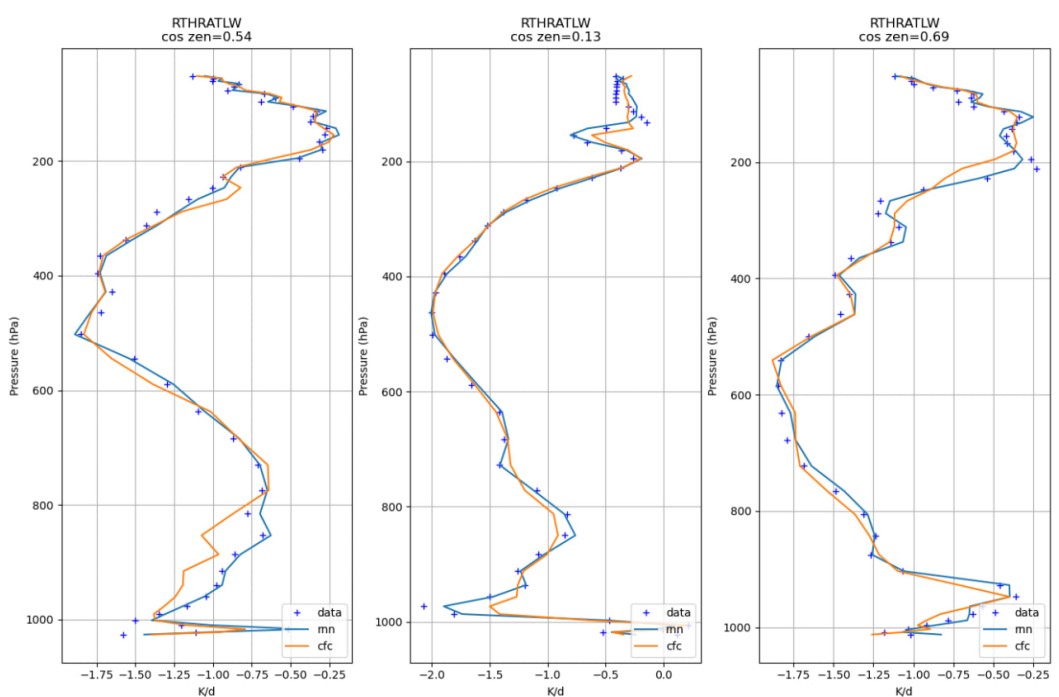

Figure 8: Exemplary long-wave profiles predicted by profile-wise RNN (blue line) and sequential CfC (orange line) models versus RRTMG data (crosses).

## B   R2 SKILLS

Fig. 9 presents mean R2 skills versus number of floating point operations. Profile-wise RNN models appear to be superior over alternative models.

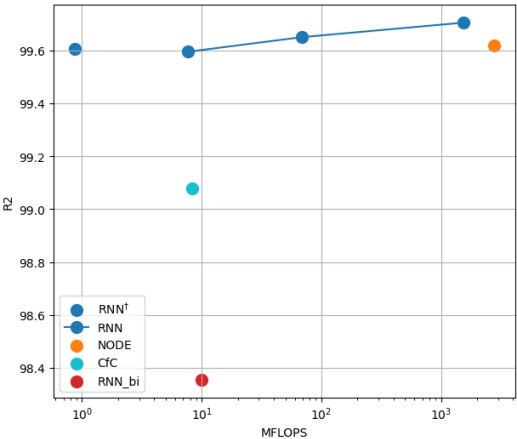

Figure 9: Mean R2 skills versus number of floating point operations. RNN indicates RNN 64,3, RNN 128,7, RNN 512,10. RNN$^\dagger$ indicates RNN 16,5

| Model | SW R2 | LW R2 |
|---|---|---|
| Multi-Output | | |
| RNN 64,3 | 99.66 | 99.53 |
| RNN 128,7 | 99.74 | 99.56 |
| RNN 512,10 | 99.80 | 99.61 |
| NODE 512 | 99.67 | 99.57 |
| CfC | 99.63 | 98.53 |
| RNN_bi | 99.10 | 97.61 |
| Single-Output | | |
| CatBoost | 95.54 | 94.24 |
| NODE | 87.51 | - |
| TMPNN | 95.64 | 96.53 |
| CNN 128 | 98.16 | 99.29 |
| CNN 256 | 98.39 | 99.37 |
| RNN 128,7 | 99.47 | 99.52 |
| RNN 256,7 | 99.48 | 99.64 |
| RNN 512,10 | 99.53 | 99.66 |
| NODE 128 | 99.31 | 99.62 |
| NODE 256 | 99.40 | 99.63 |
| NODE 512 | 99.53 | 99.67 |
| PNN 2 | 90.08 | - |
| PNN 4 | 94.05 | - |
| TMPNN 16,2 | 98.08 | 98.90 |
| TMPNN 16,7 | 99.28 | 99.11 |
| PODE | 99.31 | 98.82 |
| GRU | 98.41 | - |
| ODEGRU | 98.11 | - |
| CfC | 99.68 | 98.30 |

Table 5: R2 skills of the trained models

