# OpenReview forum: "Atmospheric Radiation Parameterization by Neural Ordinary Differential Equations and Related Models"
_ICLR.cc/2025/Conference — Submitted to ICLR 2025_

### Official Review · Reviewer_smU9 · 2024-10-16

**Soundness:** 2
**Presentation:** 2
**Contribution:** 2
**Rating:** 3
**Confidence:** 3

**Summary:**

It employs the Neuralode method to perform the Atmospheric Radiation Parameterization task.

**Strengths:**

It will be exciting to see the application of machine learning models in the field of science.

**Weaknesses:**

This work doesn't seem suited for ICLR, as it primarily appears to apply standard ML modules to AI4Science tasks. While it may perform relatively well in weather forecasting, it lacks novel insights for the ML community. The methods used are previously published and, although adapting an existing method from A to B could reach the level of a Nature/Science paper, it doesn't meet my standards for ICLR: 1 Typically, the main contribution should lie in the methodological design, incorporating modifications based on the specific problem at hand. The comparison should focus on SOTA methods to highlight its novelty. Pursuing SOTA isn't our sole objective; however, comparing only with outdated work makes it difficult to assess the novelty of the proposed method. I can't conduct all the related work and baseline studies independently.
2. If an existing method from area A is applied to area B, it should truly surprise people, as such transfers are generally not considered easy.

I believe this work falls short in both aspects.

**Questions:**

1.Why not submitting this work to a weather-related journal?
2.Could you elaborate on how you suggest the community learn from your study, beyond the application of neural ODEs in weather forecasting? Additionally, are there any recent works on this topic, aside from basic ML baseline methods discussed here? If so, please include them for comparison. If not, could you explain why? Is the field too niche, or is applying ML to this problem particularly challenging?

---

> ### Author Response · Authors · 2024-11-26
>
> Dear reviewer,
>
> We consider our work as a machine learning application for physical sciences, which falls in the scope of ICLR, stated in the call for papers.
>
> Our study shows the importance of understanding the underlying physical process to successfully emulate it with a machine learning model. The neural ode architecture was suited to emulate a fast subgrid process of radiation transfer. Through ablation study within several modifications of the architecture we were able to highlight the most important features of the model: the profile-wise iterative approach, which mimics the process of radiative flux propagation. We presume that a similar approach can be used in other tasks described by a set of integro-differential equations (in geophysics, hydrodynamics, and some other fields).
>
> Recent works studied architectures considered as SOTA, however they occur to be suboptimal, see Table 4 in the revised manuscript,  and moreover their models appear to be computationally too expensive to be implemented in operational weather forecast. While our approach gains the valuable speedup with insignificant loss in accuracy

---

### Official Review · Reviewer_tPj4 · 2024-10-19

**Soundness:** 2
**Presentation:** 1
**Contribution:** 1
**Rating:** 1
**Confidence:** 3

**Summary:**

This study uses a broad range of ML models to create emulators to represent radiative transfer in climate models. Radiative codes are one of the biggest bottlenecks in climate models. Often, radiative schemes reply on large lookup table which are hard coded into climate model fortran code. Even the most modern radiative transfer schemes do not take the full absorption band into consideration and at best rely on idealized absorption in a handful of discrete absorption bands.

The study uses a broad range of ML models, including CatBoost, Neural ODEs, CNNs, GRUs and their polynomial activation counterpart to (1) assess the performance of different architectures in emulating radiation, and (2) plug the best scheme into a regional climate model and assess speedups offered by an ML scheme as opposed to a traditional parameterization

The authors use observations in and over the Pechora Sea region for two years. One year is used for training and one year is used for testing.

More importatly, the architectures are chosen in a physics-inspired way and are categorized into three categories: point-wise (one level input used to prdict one level output), profile-wise (full vertical profile used to predict the full vertical profile) and sequential (using a sequential model like RNN to iterate and solve the radiative flux forcing at each vertical level starting from the boundaries).

The scheme with the least error is embedded into the WRF regional model and if found to provide a 4x speedup in model runtime. This testifies to the effectiveness/potential of ML models to improve traditional physical parameterizations.

**Strengths:**

The study correctly identifies the importance of radiative transfer for climate models and the fact that radiative codes form a major bottleneck in climate models. The idea that ML can solve this computational deadlock has been known for a while and some works mentioned in the study have also shown some preliminary progress in this direction.

The study has also identified a broad range of models and their appropriate application consistent with the problem domain. Single level prediction is treated as a regression and thus framework like CatBoost has been employed. Profile-wise prediction has been accomplished using multiple models including CNNs, RNNs, Neural ODEs, and polynomial networks. Lastly, inspired from vertical energy propagation, sequential algorithms like GRUs and continuous-time liquid neural nets are used for layer-by-layer prediction.

**Weaknesses:**

The study appears to fall short on multiple fronts:

1) Novelty: Apart from speedup in radiation emulation, the novelty of the work is not clear to me, since the central idea behind the work has been present by past papers and the paper pretty much uses established neural networks in their out-of-box configuration to emulate radiaton. Moreover, the authors have exhaustively used 19-20 different models of 10-11 different types. These models have not been explained in much detail (which is fine), and also the physical intuition behind simply using these many models is not clearly evident to me. If the focus is just Neural ODEs (as is the title of the submission), why not just compare it to one or two optimal baselines from past studies? I don't see any point in mixing it all up with 20 different types of models? This point is more so relevant as (to my understanding) the Neural ODE model wasn't even eventually coupled with WRF.

2) Clarity: The paper is big on breadth but not on depth. Due to this, reading this paper was a bit of a struggle as the focus has been on introducing the different architectures, and not on why they would work and others would not. Such an interpretation is likewise lacking from the results and discussion section as well. This goes back to my first point that the focus on breadth than depth makes me wonder about the novelty of the work and how it adds value to being accepted at ICLR. Most strikingly, the authors finally choose RNN 64,3 to couple to WRF, but the errors for that model have not been discussed in Table 3.

3) Lacking schematics: It would have been great if the authors would have provided detailed schematics of the main architectures used. for instance, I found figure 1 to be quite nice and I am sure I would not have been able to understand the distinction between the categories without it. For instance, what merits are obtained by choosing both ODEGRU and Liquid Neural networks, and is there solid evidence that polynomial networks might outperform relu activation based models? Or have they been used merely for intellectual curiosity?

This is nice work, but because I find the ML novelty lacking and because the ideas have already been proposed in past studies, in my opinion, the paper might better fit in a physical science journal which would allow the authors to focus on the scientific merits of their particulat analysis (which I believe may be plenty).

**Questions:**

My questions are more or less integrated with the weaknesses stated above.

---

> ### Author Response · Authors · 2024-11-26
>
> Dear reviewer,
>
> We appreciate your remarks. Please find our answer below.
>
> Comments and answers:
>
> **1. Novelty: Apart from speedup in radiation emulation, the novelty of the work is not clear to me, since the central idea behind the work has been present by past papers and the paper pretty much uses established neural networks in their out-of-box configuration to emulate radiaton.**
>
> Physical intuitions behind the use of Neural ODEs is an attempt to emulate fast subgrid processes of several transfers, scatterings and reflections of the radiation and of the overall process that settles to the observed heat and flux profiles. To support the hypothesis we studied several modifications of the Neural ODE to highlight the most important architectural features: namely, the iterative profile-wise approach. We edited the manuscript to make it more clear.
>
> **If the focus is just Neural ODEs (as is the title of the submission), why not just compare it to one or two optimal baselines from past studies?**
>
> We considered mostly the models, which are a modification of the Neural ODE. For example, the profile-wise RNN in our setting is a Neural ODE with a specific integration scheme, which  optimizes the model inference speed to make it worth as a parametrization. The out-of-box Neural ODEs are much slower than the traditional parametrization, we therefore incorporated into WRF its discrete version, RNN.
>
>
> **2. Clarity: The paper is big on breadth but not on depth. Due to this, reading this paper was a bit of a struggle as the focus has been on introducing the different architectures, and not on why they would work and others would not.**
>
> The error of the coupled model is discussed and boldfaced in Table 3. We updated our analysis in the paper to make our analysis and statements more clear.
>
>
> **3. Lacking schematics: It would have been great if the authors would have provided detailed schematics of the main architectures used.**
>
> We tried to avoid spreading the attention by including visualizations of suboptimal models. We study them in the merit of ablation study to identify the most important features of the NeuralODE-related family of architectures and its physical interpretation. Our results showed that profile-wise iterative architecture is crucial to accurately emulate subgrid processes of the radiation transfer. Polynomial activations were tried due to their relation to perturbation theory from physics (Ivanov and Ailuro, 2024), however this feature occurred to be suboptimal in the task of radiation parameterization.
>
> A. Ivanov and S. Ailuro. Tmpnn: High-order polynomial regression based on tay- lor map factorization. Proceedings of the AAAI Conference on Artificial Intelligence, 38(11): 12726–12734, Mar. 2024. doi: 10.1609/aaai.v38i11.29168. URL https://ojs.aaai.org/ index.php/AAAI/article/view/29168.

---

### Official Review · Reviewer_jGEJ · 2024-10-31

**Soundness:** 1
**Presentation:** 2
**Contribution:** 1
**Rating:** 3
**Confidence:** 3

**Summary:**

Intercomparison study of multiple neural network architectures to replace radiative transfer parametrizations inside WRF. They find an RNN to be best based on test data and then couple it to WRF via a Fortran-C++ bridge and find a 4x speed up (of the parametrization scheme) at a loss in accuracy of ~1K in surface temperature.

**Strengths:**

1) Many baselines are compared
2) The final model is coupled to the WRF model
3) Experiments should be reproducible as code is available and technical details are mentioned in the text

**Weaknesses:**

1. The loss in accuracy relative to the gained speed up seems too high. Fig. 5 and 6 give a first indication of this. The emulated parametrization can differ significantly from the original one, which would render the speed up not useful.
2. Robustness of the models is not assessed. If the proposed emulator is to be used in WRF, it need to be applicable under a wide variety of input combinations. However, this study only considers a very limited study region in high northern latitudes during only a small time period (2015-2016). Moreover, this limitation is not meaningfully discussed. As is, I suspect the parametrization would have significantly decreased skill in most real-world applications and could potentially even be catastrophically wrong (e.g. unphysical).
3. There is limited technical novelty in the contribution which makes the work less interesting to the broad readership of ICLR. More specifically, all neural network architectures studied in this work have been previously used, the dataset has been introduced in a different study (Yavich et al. 2024), the concept of emulating radiative transfer has been widely studied and the empirical results are not ground breaking.
4. Many figures could be improved: reduce white space on the sides, use consistent fonts and font sizes, think about what you want to convey with a figure and then think about how to make this message most easily accessible for the reader.
5. Many abbreviations not introduced, e.g. SW, LW
6. The tables are difficult to grasp and do not seem to have a clear message, but instead are conveying that many models have been compared and one of them turned out to be best in some cases. Also, what does bold face mean in the tables?
7. Section 2 contains lengthy descriptions of different methods, however it remains unclear how they contribute to the overall story of the paper.
8. While the study compares many baselines, it does not compare to any previously published emulators of radiative transfer schemes, which makes it difficult to assess how good the reported metrics are.
9. While reporting RMSE is important, it would be good to also evaluate additional metrics, especially those that give an indication about absolute skill levels, e.g. R^2 or relative RMSE (normalized by variability of targets).

**Questions:**

What kind of study do you intend to do with this model, that the 4x speed up in the radiative transfer parametrization scheme is necessary to make them feasible?

The main goal of this paper seems to be the speed up. However, since you only achieve a speed up at a loss in accuracy it would be important to understand this trade-off in detail. In other words, can you draw a pareto frontier showcasing the loss in accuracy vs. the gain in speed up of multiple models? Ideally with a metric that also allows to assess the accuracy of the original parametrization scheme in WRF?

---

> ### Author Response · Authors · 2024-11-26
> **Comments on weaknesses**
>
> Dear reviewer,
>
> We appreciate your remarks. Please find our answer below.
>
> Comments on weaknesses:
>
> **1. The loss in accuracy relative to the gained speed up seems too high. Fig. 5 and 6 give a first indication of this. The emulated parametrization can differ significantly from the original one, which would render the speed up not useful.**
>
> We have achieved an enhancement in accuracy in comparison to several recent publications (Table 4 in the updated manuscript). We thus expect this work to be valuable to the community.
>
> Hearting rate RMSE,K/d ---  SW   ---    LW
>
> This work    --------------------- 0.025  ---  0.111
>
> Yao et al, 2023  --------------  0.032  --- 0.139
>
> Yavich et al, 2024 ----------  0.042 --- 0.250
>
> Ukkonen, 2022 ------------- 0.160 --- n/a
>
>
> **2. Robustness of the models is not assessed. If the proposed emulator is to be used in WRF, it need to be applicable under a wide variety of input combinations.**
>
> Usually, the WRF (weather regional forecast) model is configured for each specific region. We expect that our parametrization surrogate will be trained/tuned for this region in order to achieve the best possible results on it. We focused our research on the Arctic region because of its high economic importance and complex climate. Within the Arctic region, we tested the model in different types of locations and the whole year, therefore we expect our model to be robust in all the training region. For the other region we expect that the model after retraining will give also not bad results
>
> **3.  There is limited technical novelty in the contribution which makes the work less interesting to the broad readership of ICLR.**
>
>    Technical novelty is the original profile-wise Neural ODE like architectures shown to produce the state-of-the-art results in comparison with other works (see the table above). Such an approach selection is justified by the integro-differential equation in the original model and can be applied to other similar physics problems.
>
>
> **4. Many figures could be improved**
> 	We appreciate this remark. We’ve improved the illustrations.
>
> **5.Many abbreviations not introduced, e.g. SW, LW**
>     In the updated manuscript we corrected this and other remarks.
>
> **6. The tables are difficult to grasp and do not seem to have a clear message, but instead are conveying that many models have been compared and one of them turned out to be best in some cases. Also, what does bold face mean in the tables?**
>     Boldfaced metrics are the best metrics between the compared models.
>
> **7. Section 2 contains lengthy descriptions of different methods, however it remains unclear how they contribute to the overall story of the paper.**
>
> We studied various  Neural ODE models to identify the most important elements of the architecture. We found that an iterative profile-wise architecture is necessary to emulate radiative flux propagation.
>
> We studied them as an ablation study to find the essential part of the architecture: the profile-wise approach and iterative architecture to emulate the subgrid process of heat profile settling. We’ve edited the manuscript to make it more clear.
>
> **8. While the study compares many baselines, it does not compare to any previously published emulators of radiative transfer schemes, which makes it difficult to assess how good the reported metrics are.**
>
> See Table above. We added this Table to the manuscript with the remark that our region and atmosphere model were different from those studied in these articles.
>
>
> **9. While reporting RMSE is important, it would be good to also evaluate additional metrics, especially those that give an indication about absolute skill levels, e.g. R^2 or relative RMSE (normalized by variability of targets).**
>
> We appreciate this remark. We have added Appendix B with R2 skills. For our best models, it is about 99.7%
>
>
>
> Ukkonen. Exploring pathways to more accurate machine learning emulation of atmospheric
> radiative transfer.Journal of Advances in Modeling Earth Systems, 14(4), April 2022.
> doi:10.1029/2021ms002875. URL https://doi.org/10.1029/2021ms002875.
>
> Yao et al, A physics-incorporated deep
> learning framework for parameterization of atmospheric radiative transfer.
> Journal of Advances in Modeling Earth Systems, 15(5), 2023. doi: 10.1029/2022ms003445. URL https:
> //doi.org/10.1029/2022ms003445.
>
> Yavich et al, A physics-inspired neural network for short-wave radiation parameterization.
> Journal of Inverse and Ill-posed Problems, 2024. doi: 10.1515/jiip-2023-0075. URL https:
> //doi.org/10.1515/jiip-2023-0075.

---

> > ### Comment · Reviewer_jGEJ · 2024-11-28
> >
> > Dear authors,
> >
> > thanks a lot for responding to my concerns.
> > As you write yourself: "a direct comparison of the numbers in Table 4 is injudicious" - which, along with the other replies and the fact that you are still conducting new experiments, leaves me with the impression that this work can benefit a lot from another round of improvements.
> > I will not raise my score, but I would nonetheless very much like to encourage you to continue to refine this work and, when further improved, to submit to a Journal.
> >
> > All the best,
> > Reviewer jGEJ

---

> ### Author Response · Authors · 2024-11-26
> **answers to the questions**
>
> **What kind of study do you intend to do with this model, that the 4x speed up in the radiative transfer parametrization scheme is necessary to make them feasible?**
>
> Firstly, We added a new model to our experiments that ultimately gave a speed up 26.5x, see update manuscript.
>
> Secondly, radiative transfer is a bottleneck in atmospheric modelling, therefore speeding it up dramatically speeds up the whole atmospheric modelling. The figure 5b clearly shows the overall speedup of the atmospheric computations. We received a speedup of 1.5 times .
> This means
> * 1.5 times faster operational weather forecast, what is quite important since numerical weather prediction typically takes many hours of computational time.
> * 1.5 times faster studying of extreme weather scenarios, which is quite important since dozens of different scenarios should be evaluated.
> * less carbon footprint.
> Yet we admit there is a gap of improvement in our results. In particular, our linking scheme of the inference routine to WRF evidently produces some overhead. We plan to study other approaches in the future.
>
> **accuracy vs. the gain in speed up of multiple models**
>
> We added in Appendix B a plot showing performance of different models in (R2, MFlops) coordinates.

---

### Official Review · Reviewer_DThb · 2024-11-03

**Soundness:** 3
**Presentation:** 3
**Contribution:** 2
**Rating:** 5
**Confidence:** 3

**Summary:**

This study presents the application of neural networks, particularly Neural ODEs, for surrogate radiation parameterization. It aims to reduce the runtime of radiation parameterization in numerical weather forecasting and enhance overall forecasting efficiency by leveraging the rapid inference capabilities of neural networks. To achieve this, the study approximates the heat transfer process using an ODE with an unknown right-hand side, which is estimated by a neural network. The experiments evaluated the practicality of 25 models across three types, integrating the machine learning model with the real WRF numerical model. Compared to traditional parameterization schemes, the module's runtime improved fourfold, while maintaining consistent prediction accuracy.

**Strengths:**

1. This work explores the application of a series of neural networks, primarily Neural ODEs, in radiation parameterization, and incorporates RNNs into the WRF model for predictive experiments, achieving practical testing of the neural network parameterization scheme.
2. The manuscript is clearly written, adheres to formatting requirements, and contains no obvious LaTeX errors.
3. The algorithm design follows the physical principles of radiation parameterization, successfully integrating data-driven approaches with physical information.

**Weaknesses:**

1. **Method Description**: The explanation of the method selection could be more thorough. It would be beneficial for the authors to elaborate on the specific advantages of Neural ODE in its application for this radiation parameterization task compared to other ML methods. Additionally, I encourage them to clarify how the parameterization substitution task differs from standard regression and ODE fitting tasks in terms of unique challenges or requirements. It might also be worth considering why well-known and high-performing architectures like neural operators and ResNet were not included as baselines in this study. I encourage the authors to explain their rationale for selecting the baseline models they used and to clarify whether they considered including these popular architectures.

2. **Experimental Analysis**: The experimental analysis would benefit from further clarity and depth. Although the study evaluates 11 architectures across three categories and compares their accuracy, further exploration of the results is necessary for a complete understanding. I encourage the authors to provide a more detailed analysis of the performance differences between architectures, particularly for the RNN and Neural ODE models in shortwave and longwave predictions. Additionally, a discussion of potential reasons for these differences based on the characteristics of each architecture and the nature of the prediction tasks would greatly enhance the overall analysis.

3. **Significance**: In the final WRF joint experiment, the choice to use a profile-wise convolutional RNN instead of Neural ODE raises an important question about the significance of the findings. It would be helpful for the authors to clarify their reasoning for selecting the RNN model for this experiment, particularly given that Neural ODE is discussed as a key focus of the paper. If the choice of RNN, is primarily to demonstrate that profile-wise machine learning parameterization schemes (like Neural ODE and RNN) are superior, this seems less novel, as prior work[1-3] has already established similar conclusions without framing them as profile-wise methods.

[1] Wang, Jiali, Prasanna Balaprakash, and Rao Kotamarthi. "Fast domain-aware neural network emulation of a planetary boundary layer parameterization in a numerical weather forecast model." _Geoscientific Model Development_ 12.10 (2019): 4261-4274.

[2] Jakhar, Karan, et al. "Learning closed‐form equations for subgrid‐scale closures from high‐fidelity data: Promises and challenges." _Journal of Advances in Modeling Earth Systems_ 16.7 (2024): e2023MS003874.

[3] Eyring, Veronika, et al. "Pushing the frontiers in climate modelling and analysis with machine learning." _Nature Climate Change_ (2024): 1-13.

**Questions:**

1. Would it be possible to provide a clearer theoretical justification or explanation for the viewpoint that the profile-wise convolutional RNN is a discrete substitution of the Neural ODE? What are the advantages of Neural ODEs when handling regular (discrete) time data?

2. Given that GRUs are an enhancement of RNNs, could you clarify why they are not suitable for profile-wise type experiments?

3. In lines 221 to 222, could you provide a more detailed explanation of the "Spatial grid"? I noticed that Figure 2 only illustrates random location nodes.

4. Regarding the selection of the test set mentioned in line 237, is there a temporal uniformity in this selection? Considering the variations in day-night cycles and seasonal changes in polar regions (line 223), how might this impact the selection?

---

> ### Author Response · Authors · 2024-11-26
> **comments on weaknesses**
>
> Dear reviewer,
>
> We appreciate your remarks. Please find our answer below.
>
> Comments on weaknesses:
>
> **1. Method Description: The explanation of the method selection could be more thorough.**
>
> The following text was added to the manuscript.
>
> Atmospheric radiative heating is considered as an adiabatic process in weather forecast and climate modelling. Radiation transfer itself is described at the quantum level  and involves numerous transfers, scatterings and reflections. We found that an iterative profile-wise architecture for a Neural ODE is necessary to emulate radiative flux propagation and the underlying physics. On the other hand, sequential and shallow networks architectures emulate radiation transfer in a single direction only, therefore are inferior, which is supported by our experiments.
>
>
> **2. Additionally, I encourage them to clarify how the parameterization substitution task differs from standard regression and ODE fitting tasks in terms of unique challenges or requirements.**
>
> We were looking for a regression model that would not only fit the data, but also feature the underlying physics. This is related to the fact that radiative transfer is governed by an integro-differential equation (Yavich et al, 2024) rather than an ODE. Consequently, special attention was given to architectures that process data as a whole sequence completed with multi-step evolution.
>
> **3. It might also be worth considering why well-known and high-performing architectures like neural operators and ResNet were not included as baselines in this study.**
>
> The recent work of Yao et al, 2023 tested FNO, ResNet, and transformers for this task. However, none of them were found optimal. We thus did not include these architectures in our study.
>
>
> **4. Experimental Analysis: The experimental analysis would benefit from further clarity and depth.**
>
> We consider most of the tested architectures as modifications of neural ode architecture to ablate and figure out the essential architectural features for the task of emulating radiation transfer. The results support our hypothesis that RNN iterative profile-wise architecture is necessary to emulate radiative flux propagation. We improved the discussion in order to address this question better
>
>
> **5. Significance: In the final WRF joint experiment, the choice to use a profile-wise convolutional RNN instead of Neural ODE raises an important question about the significance of the findings.**
>
> We consider a very specific RNN model as a neural ODE with a special integration scheme, which is aimed at reducing computational complexity without losing crucial features. Our RNN involves iterative profile-wise architecture to emulate radiative flux propagation. Hence, we do not see a contradiction here.
>
> Yao et al, A physics-incorporated deep learning framework for parameterization of atmospheric radiative transfer.
> Journal of Advances in Modeling Earth Systems, 15(5), 2023. doi: 10.1029/2022ms003445. URL https:
> //doi.org/10.1029/2022ms003445.

---

> ### Author Response · Authors · 2024-11-26
> **answers to the questions**
>
> Answers to the questions:
>
> **1. Would it be possible to provide a clearer theoretical justification or explanation for the viewpoint that the profile-wise convolutional RNN is a discrete substitution of the Neural ODE? What are the advantages of Neural ODEs when handling regular (discrete) time data?**
>
> Thank you for a good question!
> Firstly, Neural ODE is always in practice discretized with the help of some numerical scheme. Imagine the simplest of the possible schemes - the explicit Euler, where $x(t+\Delta t) = x(t) + f(x(t),t)\cdot \Delta t$. It is identical by construction to the RNN we used and the neural network we use in RNN is the same CNN we use for parametrizing $f(x,t)$. We improved the section 2.4 SUBSTITUTING NEURAL ODE in order to make the analogy more obvious for readers.
>
> As for the second question - Neural ODE has shown practically the same results as an analogous RNN, however if we know the process is captured by continuous nonlinear dynamics, sometimes NeuralODE gives more accurate results because of smaller time step. It’s worth noting that our time is virtual time, it has the physical meaning of the variable describing the evolution of multiple radiation reflections and transmissions and the establishment of the final radiation profile. In our case the process can be well described by the iterative scheme which is proven by good correspondence between RNN and Neural ODE performance. However Neural ODE gives us valuable insights about this possible RNN substitution and helped us to obtain the final computation-effective solution.
>
> In a neural ODE, one can choose the forward Euler scheme as an ODE solver. Each step of the scheme then can be considered as a feed-forward network with residual connection. To make several steps, one should reapply the same network several times, therefore it can be considered as an RNN, where the hidden state is propagating without assimilation of any input. A more thorough proof can be found in Ivanov and Ailuro, 2024 or other publications.
>
> **2. Given that GRUs are an enhancement of RNNs, could you clarify why they are not suitable for profile-wise type experiments?**
>
> GRU model is suitable and can be inferred as a specific gated version of the RNN. However, the common GRU feature of selective input assimilation is completely lost in our profile-wise approach emulating the Neural ODE. Probably, an  interesting substitution would be to try some attention mechanism to use only the important parts of the previous profile, however it would strongly increase the computational time so we skipped it.
>
>
> **3. In lines 221 to 222, could you provide a more detailed explanation of the "Spatial grid"? I noticed that Figure 2 only illustrates random location nodes.**
>
> WRF uses the 3D spatial grid for numerical solution of the nonlinear partial-differential equations describing the dynamics of the atmosphere (thermodynamics, moisture convection and others). We picked a small fraction of the modelled data for ML models learning and testing (shown as points in Fig. 2).
>
>
> **4. Regarding the selection of the test set mentioned in line 237, is there a temporal uniformity in this selection?**
>
> Training and testing data involved uniformly distributed data within a year: 350 days for testing (essentially all the days of 2015) and 91 days for training (every fourth day of 2016). Every chosen day included data within 24 hours (sampled every hour) and at 100 random surface locations. This type of data filtration made the dataset computationally feasible and removed redundant information, while keeping sufficient representation of day/night and seasonal changes.
>
> A. Ivanov and S. Ailuro. Tmpnn: High-order polynomial regression based on Taylor map factorization. Proceedings of the AAAI Conference on Artificial Intelligence, 38(11): 12726–12734, Mar. 2024. doi: 10.1609/aaai.v38i11.29168. URL https://ojs.aaai.org/ index.php/AAAI/article/view/29168.

---

> ### Comment · Reviewer_DThb · 2024-12-03
> **Reply to Authors' Revisions**
>
> Thank you for your thoughtful responses to my questions. The added content has certainly made the paper clearer, particularly the comparison between Neural ODEs and RNNs. I will not change my score, but this does not mean the work lacks significance. It simply suggests that for applied papers, it is crucial to highlight the irreplaceability of the chosen methods. I recommend that you reorganize the paper following this approach and consider submitting it to other high-level conferences or journals.

---

### Author Response · Authors · 2024-11-26

Dear reviewers,

 We updated the manuscript to accommodate a new model and several experiments.
 The new text is in red for your convenience.

 If there are any further questions or points that need discussion, we will be happy to address them.

 Best regards,

 The Authors

---

### Meta-Review · Area_Chair_6hjQ · 2024-12-19

**Metareview:**

The authors use neural ODEs as a surrogate model for radiation parameterization with the goal of reducing runtime of weather forecasting models where the radiation process take up significant time. They integrate their surrogate model with the numerical model (WRF) in a specific region and show good speedups (1.5x) for the weather forecasting. They also benchmark many ML models and show their model performs best.

Strengths: This work extends previous works in using surrogates for the radiation parameterization component of weather models to speed  up forecasting - this can have important implications to future weather models. Their neural ODE formulation is an interesting contribution and they also show this ML model is more performant than previous models.

Weaknesses: the speedup gains come at the cost of accuracy - there is limited evaluation on how useful this speedup is, robustness of the model in different regions is not assessed, and the paper is broad with evaluations on many architectures with minimal analysis on why so many architectures were evaluated, limiting the overall depth of the contributions

The following improvements could strengthen the paper: detailed analysis on the cost vs accuracy tradeoffs given that the use of a surrogate model in itself is not a new contribution, testing the models in different regions (and possibly demonstrating OOD effects), and repeating the above for a smaller set of strong baselines that are motivated carefully in relation to the problem that is being solved.

**Additional Comments On Reviewer Discussion:**

The reviewers primarily raised weaknesses described above. Reviewer smU9 also indicated the paper would benefit from a different venue and ICLR is not the right place - this has not factored into my final review and I believe AI for science methods have a place in ICLR as well.

Reviewer DThb highlights "This does not mean the work lacks significance. It simply suggests that for applied papers, it is crucial to highlight the irreplaceability of the chosen methods." This was factored into the my suggestions for improvements.

There were other concerns highlighted regarding the presentation that the authors have addressed in their rebuttal.

---

### Decision · Program_Chairs · 2025-01-22

Reject